# Macro tax burden, FDI, and national innovation efficiency: A study on the impact of macro tax burden on national innovation efficiency

Xiaobo Shen[1], Pingsheng Dai[2]*

1 School of Economics, Xiamen University, Xiamen, Fujian, China, 2 School of Finance and Economics, Jimei University, Xiamen, Fujian, China

* daips@jmu.edu.cn

**Data Availability Statement:** The datasets generated and/or analyzed during the current study inclued in this article can be accessed from the WDI and WGI databases of the World Bank and the

## Abstract

Based on a panel dataset of 54 countries from 2008 to 2019, this article uses the mediation effect model to examine the relationship between macro tax burden, FDI and innovation efficiency. We find that:(i) the macro tax burden is positively correlated with the innovation efficiency; (ii) there is a non-linear effect of FDI on innovation efficiency conditional on macro tax rate. When the macro tax burden is greater than the critical value (25.28%), it indirectly limits innovation efficiency by hindering FDI inflows. This means that in order to promote innovation efficiency at the national level, the macro tax rate should be maintained at a reasonable level, because that can make the government raise more money to invest in and subsidy the innovation activities.

## Introduction

Since the 90s of the last century, in order to implement the strategic goal of innovative economy, many countries have introduced preferential tax policies to promote enterprise innovation. For instance, the US government has twice enacted the R&D tax credit system [1], the Canadian government has introduced federal and provincial R&D credit programs [2], and the Chinese government has adopted financial incentives such as direct subsidies and tax breaks [3]. FDI is often attracted by countries identified as having a successful innovation performance [4], and the available literature suggests that FDI can benefit the innovation activities of host countries due to the large amount of technology and experience transfer of MNEs [5, 6]. Under this background, researchers pay much attention to whether the macro tax burden of host countries affects FDI's spillovers. A too low tax burden not only weakens the government revenue, but also dilutes innovation incentives of the government's tax credit policies. According to Shapir-Tidhar et al. [7], in the case of high tax burden (tax rate >25%), a decentralized tax structure can reduce people's tax burden perception and improve the efficiency of tax administration. When the overall tax burden is high, the tax credit, like the decentralization of taxation, can reduce the tax burden perception of enterprises and encourage

HDR database of the United Nations Development Programme. (https://databank.worldbank.org/; https://www.undp.org/).

**Funding:** The author(s) received no specific funding for this work.

**Competing interests:** The authors have declared that no competing interests exist.

them to actively participate in innovation activities; while the overall tax burden is low, the effect of tax credit in reducing the perception of tax burden is weakened, and the incentive effect of tax credit policy on participation in innovation activities is diluted. Therefore, developing innovative economy has put forward new requirements for the macro tax rate of host countries. Then, whether does macro tax burden affect the national innovation efficiencies? Especially, if does it affect the innovation efficiencies through FDI's spillovers? Is this effect non-linear? Can governments design reasonable macro tax rates to maximize the FDI's spillovers? Although the above issues have been studied or addressed in the existing literature, there is no consensus among researchers. However, finding solutions to these questions is very important since it can help governments to design pertinent macro tax rates, and help keeping its attractions to FDI and accelerating technological innovations, and providing a long-run effective driver for innovation-driven economy.

Based on the mediation effect and the threshold effect models, this paper examines the effect of the macro tax burden of host countries on their innovation efficiencies by using a panel dataset from 54 countries from 2008 to 2019. Following Gonzalez-Blanco et al. [8], we use the technological efficiency of innovation activity as a measurement of innovation efficiency (INN). We find that:(i) the macro tax burden is positively correlated with the innovation efficiency; (ii) there is a non-linear effect of FDI on innovation efficiency conditional on macro tax rate. When the macro tax burden is greater than the critical value (25.28%), it indirectly limits innovation efficiency by hindering FDI inflows. The findings of this paper have obvious theoretical and policy implications. In terms of theoretical implication, this paper finds that there is a nonlinear relationship between macro tax burden and innovation efficiency. In terms of policy implications, the results of this paper imply that the macro tax burden should be maintained at an appropriate level to prevent excessive tax burdens from inhibiting FDI inflows, thereby hindering innovation efficiency.

The contribution of this paper is reflected in two aspects. First, this paper constructs a structural equation model (SEM) with threshold effect to investigate the nonlinear relationship between macro tax burden, FDI and innovation efficiency. We find that, in general, the macro tax burden is positively correlated with innovation efficiency, but an excessively high macro tax burden will hinder FDI inflows, which will adversely affect innovation efficiency. Second, this paper explicitly examines the threshold effect of macro tax burden and the mediating effect of FDI, which are two issues that are relatively neglected in the existing literature. We hope that these findings will enrich the existing literature on the relationship between macro tax burden, FDI and innovation efficiency. Of course, although structural equation modeling is a popular data analysis tool, its limitations also require attention. For example, when the sample size is too small, there will be an overfitting problem, making the estimation untrustworthy. In this paper, a panel dataset of 54 countries from 2008 to 2019 is used to estimate our SEM, and the sufficient sample size can promise the credibility of the estimations.

According to the theory of the Keynesian school, tax reduction can mobilize the enthusiasm of investment, stimulate the economic vitality of the market, therefore it can be a steady driver of economic growth [9]. From the perspective of welfare economics, the basic functions of taxation include maintaining a certain tax burden, expanding the supply of public goods, and improving the level of social welfare [10]. On the other hand, tax increase can expand fiscal revenues and public investment to improve the infrastructure and the environment of life and production, thus helping the flow of FDI [11]. As a result, there is a dilemma between maintaining (even raising) tax rates and decreasing tax rates.

Regarding the relationship between FDI and taxes, some authors think that a lower tax burden can attract foreign enterprises to investment and expand the scale of FDI, accelerating the

development of the host country's economy [12–14]. However, some authors suggest that tax burden does not have a significant effect on FDI inflows in a favorable institutional environment [15, 16]. The above-mentioned literature shows that the relationship between tax burden and FDI is not simply linear, and there may be a nonlinear impact of tax burden on the spillovers of FDI.

Many researchers theoretically examine the channels of the FDI's spillovers. Cheung and Lin [5] and Lin and Lin [17] think that FDI can generate innovation spillovers by several channels. First, local enterprises can learn foreign technologies and produce similar products by learning and doing. Second, local enterprises can obtain foreign skills by the moving of the skilled-workers employed the FDI enterprises. Third, inward FDI can inspire and incentivize local enterprises by demonstration effect, resulting in the shortage in the process of trials-and-errors. Fourth, FDI enterprises can affect local suppliers' innovative activities by technologies transfer and employees training. Fifth, FDI can supplement the investment expenditure of host countries, including local innovative investments. Sixth, FDI can impose pressure on local enterprises due to intense competitions caused by the advanced techniques and higher productivity of FDI enterprises, promoting local enterprises to increase R&D investment and seek innovation improvement [5, 17–20].

Some empirical studies confirm that FDI has a positive impact on innovation [18, 19, 21–23]. However, there is evidence for some exceptions in literature. For example, Stiebale and Reize [24] find that FDI through merger and acquisition (M&A) has a negative impact on R&D inputs, and that M&A has no significant effect on innovation output. Goel and Saunoris [20] show that FDI strengthens the number of non-innovators in host countries and weakens the innovation output of host countries. Meanwhile, FDI has a crowding out effect on host countries' innovation spillovers, and local firms with insufficient R&D investment or no sunk research costs may be more likely to abandon innovation pursuits and choose the role of imitators in the face of greater FDI [25].

Additionally, empirical literature find that the spillovers of FDI is affected by the absorption capacity of host countries, and factors such as institutional environment and labor characteristics (age, education level, etc.) may influence the spillovers of FDI [19].

Some studies reveal that the tax burden has a negative effect on innovation. Stantcheva [26] finds that taxes can reduce the expected net return on innovation inputs and have an impact on the quantity, quality and location of innovation. Akcigit et al [27] examine the impact of corporate taxes on innovation activity in the United States during the 20th century. They find that a higher corporate income tax negatively affects the corporates' inventive activities. Faber et al. [28] find that the corporate tax burden negatively contributes to the innovation output in the OECD countries. However, there are some empirical literature that find the tax burden can promote the corporate innovation activity. For example, Nam [29] finds the after-tax NPV of corporate will increase with the corporate tax rate, as a result, a "tax-rate-increase-cum-base-broadening" policy may provide a stronger driver for the corporate R&D expenditure. Furman et al. [30] think that the national innovative capacity is the ability of a country in producing novel technologies and commercializing them in the long term, which depends on the public innovation infrastructure and the environment for a nation's industrial clusters innovation. Since the higher a nation's macro tax burden is, the more the government revenues are, so the government can invest more money in the public innovation infrastructure, which can in turn promote the innovation efficiency of enterprises. Based on the framework of Furman et al. [30], Hu and Mathews [31] concludes the same conclusion for five latecomers of East Asian economies. According to the above mentioned literature, the macro tax burden has both negative and positive effects on innovation, so the net effect depends on which one of the two opposite effects is greater.

The existing studies discuss the relationship among tax burden, *FDI*, and innovation. A low tax burden is conducive to reducing costs and maintaining corporate profit space, maintaining the attraction of foreign direct investment, reducing financial pressure on corporate R&D investment, and promoting technological innovation and industrial upgrading. In contrast, a high tax burden may squeeze the space for investment returns and hinder the inflow of *FDI* and its innovation spillover effects, thus, generating negative impact on innovation efficiencies.

This paper is mainly concerned with two issues, one is the direct effect of macro tax burden on innovation efficiency, and the other is how macro tax burden indirectly affects innovation efficiency by affecting *FDI* inflows. In order to answer these two questions, this paper examines the direct effect of macro tax burden on innovation efficiency and the indirect impact through FDI based on the mediation effect method, and then examines how the impact of *FDI* inflow on innovation efficiency changes with the macro tax burden based on the panel threshold model. We believe that examining the above two issues can provide insights for governments to formulate appropriate macro tax policies to attract *FDI* inflows and promote innovation effects.

## Materials and methods

### Ethical approval

Ethical approval was not required as the study did not involve human participants.

**Models.** *The mediation effect model.* This paper uses a three-step mediation effect model [32] to examine the relationship between macro tax burden, FDI and technological innovation efficiency. In the first step, a regression model is used to test the linear impact of macro tax burden on technological innovation. In the second step, a regression model is applied to identify the impact of macro tax burden on FDI. In the third step, a regression model is utilized to test the impact of macro tax burden and FDI on technological innovation. The regression equations of the above three steps constitute a structural equation model (*SEM*) to identify the effect of macro tax burden on innovation efficiencies, especially its effect through FDI. Theoretically, the macro tax burden can affect the innovation efficiency by affecting FDI inflow, because a high macro tax burden will reduce the after-tax return of FDI, thus inhibiting the inflow of FDI, limiting the spillover effect of FDI, and ultimately negatively affecting the innovation efficiency. Therefore, we construct the following mediation effect model to explore the mediating effect of FDI:

$$
\begin{aligned}
INN_{it} &= \alpha tax_{it} + \theta_n C_{it}^n + \epsilon_{it} \\
FDI_{it} &= \beta tax_{it} + \theta_n C_{it}^n + \epsilon_{it} \\
INN_{it} &= \lambda FDI_{it} + \alpha' tax_{it} + \theta_n C_{it}^n + \epsilon_{it}
\end{aligned}
\tag{1}
$$

where $INN_{it}$ is the dependent variable, indicating the innovation efficiency of the country *i* in the year *t*. The variable tax represents the tax burden, and the variable *FDI* is the level of foreign direct investment. $C^n$ is a vector composed of n control variables. $\alpha$, $\beta$, $\lambda$, and $\alpha'$ are parameters to be estimated, $\theta_n$ is a vector composed of the estimated coefficients of n control variables. The error term $\epsilon_{it}$ $N(0, \sigma^2)$.

*The mediation effect model with threshold effect.* In the above model (1), there is linear relationship between *FDI* and *INN*. However, conditional on the macro tax burden, the relationship between *FDI* and *INN* may change. Specifically, when macro tax burden is lower, *FDI* may bring about significant spillovers, thereby promoting innovation efficiencies; while macro

tax burden is high, inward *FDI* may be hindered, thereby negatively affected innovation efficiencies. To test the non-linear relationship between *FDI* and innovation conditional on the macro tax burden, based on Hansen [33], we specify the following mediation effect model with threshold effect:

$$
\begin{aligned}
INN_{it} &= \alpha tax_{it} + \theta_n C_{it}^n + \epsilon_{it} \\
FDI_{it} &= \beta tax_{it} + \theta_n C_{it}^n + \epsilon_{it} \\
INN_{it} &= \lambda_1 FDI_{it} I(tax_{it} \leq \gamma_1) + \lambda_2 FDI_{it} I(tax_{it} > \gamma_1) + \alpha' tax_{it} + \theta_n C_{it}^n + \epsilon_{it}
\end{aligned}
\tag{2}
$$

where, the variable *tax* is a threshold variable, the $\gamma_s$ are the possible critical values of the threshold variable tax, $I(\cdot)$ is an indicator function. When one of the conditions in the brackets of the indicator function is met, $I(\cdot) = 1$, otherwise 0.

**Variables and data.** *Variables*. (1) Innovation efficiency index (*INN*). The dependent variable of models (1) and (2) is the innovation efficiency index of a country. In earlier literature, the number of patents is used as a proxy of innovation. However, the number of patents only reflects the output of a country's innovation activity, it cannot reveal the performance of a country to transfer its innovation inputs into the innovation outputs. This paper use the technological efficiency of innovation activity as the measure of innovation efficiency index. In order to measure the technological efficiency of a nation's innovation, we specify the following translog production function for innovation activity:

$$
lnY_{it} = \beta_0 + \beta_1 lnK_{it} + \beta_2 lnL_{it} + \frac{1}{2}\beta_3 (lnK_{it})^2 + \frac{1}{2}\beta_4 (lnL_{it})^2 + \beta_5 lnK_{it}L_{it} + v_{it} - \mu_{it}
\tag{3}
$$

where, the variable $L_{it}$ is the labor input in the innovation activities of country *i* in the year *t*, $K_{it}$ is the *R&D* expenditure of country *i* in the year *t*, and $Y_{it}$ is the output of innovation activities of country *i* in the year *t*, measured as the number of granted patents. $v_{it}$ is a random error term, we suppose $v_{it} \sim N(0, \sigma^2)$. $\mu_{it}$ is inefficiency term, and we suppose $\mu_{it} \sim N_+(\mu_{it}, \sigma^2)$. We can estimate the coefficients of Eq (3) using stochastic frontier analysis (*SFA*), then calculate the innovation efficiency index (*INN*) according to the estimated coefficients and inefficiency values.

(2) Macro tax burden (*tax*). The core independent variable of model (1) is the macro tax burden, at the same time, this variable acts as the threshold variable in model (2). The variable is measured as the ratio of total tax revenues to *GDP*.

(3) Foreign direct investment (*FDI*). The variable *FDI* is the mediator variable of model (1). It is measured as the proportion of *FDI* inflow to *GDP*.

(4) The relevant control variables. Based on relevant literature, the control variables of the mediator effect model (1) include the following factors:

*a.* Human capital (*edu*). The variable *edu* is represented by the education expenditure to *GDP*. Successful innovation critically depends on the knowledge, skills, and capabilities of the members of an organization, so human capital is an important driver of innovation [34]. Most empirical studies support a positive correlation between human capital and innovation [35–37], though some studies suggest that the impact of human capital on innovation is not significant or negatively affected [38–40].

*b.* Internet penetration (*inter*). The variable inter is measured as the ratio of internet users to population. The use of the Internet can facilitate the generation and dissemination of knowledge [41, 42]. Empirical studies have shown that Internet access reduces the cost of information dissemination and leads to more patenting outcomes [43–45], but Xiong

et al [45] find that the Internet has an inverted *U*-shaped effect on innovation output, that is, Internet penetration above a certain level will reverse the constraint of innovation output.

*c*. Industrialization (*indu*). The variable *indu* is defined as the proportion of value-added of the secondary industry to *GDP*. Innovation has been a key source of competitive advantage since the beginning of the Industrial Revolution [46]. Numerous empirical studies have shown that firms that successfully leverage innovation strategies gain a range of benefits in achieving higher margins and more market share, with industrialization underscoring the importance of technology and R&D for innovative activity [47, 48].

*e*. Urbanization (*urba*). The variable urba is calculated as the proportion of people living in city areas. There is a spatial aggregation of innovation activity, and patent sources tend to accumulate in large cities [49, 50]. Many empirical studies support that urbanization favors the innovation output of countries [37, 51, 52].

*f*. Trade openness (*trad*). The variable trad is measured as the ratio of import and export to *GDP*. Driven by the open economy, opening up to other countries can bring more innovative resources and impetus. Lin and Lin [17] have shown that imports will intensify market competition, promote local enterprise innovation to maintain market share by improving efficiency. Under the pressure of international competition, exporting enterprises need to continue to innovate in order to gain shares and advantages in the international market. Therefore, import and export has a positive impact on the innovation activities of enterprises.

*g*. Institutional environment (*WGI*). The variable *WGI* is represented by the normalized world governance index from the World Bank, which averages six indicators of voice and accountability, political stability and no violence, government effectiveness, regulatory quality, rule of law, and control of corruption. A good institutional environment is considered as one of important factors in attracting more FDI inflow, thus facilitating innovation output. However, recent empirical studies provide interesting results: Xiong et al. [45] show that the political stability of countries has a significant negative impact on innovation output, and political corruption is conducive to innovation output, and only government effectiveness has a positive impact on innovation output. By data analysis of the former Soviet Union, Aghazada and Ashyrov [53] concluded that there is a positive correlation between bribery and corporate innovation.

*h*. The level of economic development (*GNIpc*). The variable *GNIpc* is the *GNIpercapita*. It is used as a proxy variable for the economic development level of each country. Financial constraints and lack of demand are major barriers to corporate innovation activity [54–56], a higher economic development level is helpful to overcome financial constraints on innovation inputs and generate more demand for innovation.

*Data*. The data on the relevant variables are obtained from the WDI and WGI databases of the World Bank and the HDR database of the United Nations Development Programme. Specifically, the data on GNI per capita are taken from the HDR database, and the data on the remaining variables are taken from the WDI and WGI databases. For some countries and some years, there are missing values in terms of the tax revenues in the WDI database. We use the mean interpolation to fill those missing values. The time window of our sample data spans 2008- 2019 with 54 sample countries (see Table 1). Table 2 reports the descriptive statistics of the related variables.

**Table 1. List of 54 countries.**

| Country | | | Country | | |
|---|---|---|---|---|---|
| Argentina | Colombia | Hungary | Luxembourg | Portugal | Sri Lanka |
| Australia | Costa Rica | India | Malaysia | Romania | Switzerland |
| Austria | Croatia | Indonesia | Malta | Russian | Thailand |
| Belgium | Cyprus | Iran | Mauritius | Serbia | Tunisia |
| Brazil | Czechia | Ireland | Mexico | Singapore | Turkiye |
| Bulgaria | Egypt | Italy | Netherlands | Slovak | Ukraine |
| Canada | France | Jordan | New Zealand | Slovenia | United Arab |
| Chile | Germany | Kazakhstan | Norway | South Africa | United Kingdom |
| China | Greece | South Korea | Poland | Spain | United States |

**Table 2. Descriptive statistics of the relevant variables.**

| Variable | Definition of variable | Obs. | Max. | Min. | Mean | S.E. |
|---|---|---|---|---|---|---|
| INN | Innovation efficiency | 648 | 0.7455 Costa Rica | 0.2139 Ireland | 0.5528 | 0.1118 |
| FDI | Proportion of FDI inflow to GDP | 648 | 2.7935 Cyprus | 0.0000 Italy | 0.0847 | 0.2377 |
| tax | Proportion of macro tax to GDP | 648 | 0.4605 Cyprus | 0.0004 United Arab | 0.1725 | 0.0586 |
| edu | Proportion of public education expenditure to GDP | 648 | 8.0306 Norway | 1.3052 United Arab | 4.6441 | 1.2230 |
| inter | Internet penetration rate | 648 | 0.9915 United Arab | 0.0438 India | 0.6303 | 0.2276 |
| indu | Proportion of secondary industry value- added to GDP | 648 | 0.5804 United Arab | 0.0997 Cyprus | 0.2685 | 0.0818 |
| urba | Proportion of urban residents to total population | 648 | 1 Singapore | 0.1820 Sri Lanka | 0.7062 | 0.1664 |
| trad | Proportion of foreign trade to GDP | 648 | 4.3733 Singapore | 0.2211 Brazil | 0.9950 | 0.6981 |
| WGI | World governance index | 648 | 1 Norway | 0 Egypt | 0.5627 | 0.2790 |
| GNIpc | Logged gross national income per capita (PPP US dollars) | 648 | 11.3861 Singapore | 8.2156 India | 10.1773 | 0.6377 |

# Results

## The results of the mediation effect model (1)

We first estimate SEM model (1) with the country fixed and the year fixed effects. The columns (1) and (2) of Table 3 display the estimated results of SEM model (1). Column (1) displays the coefficients of the first equation of model (1). The estimated coefficient of the variable tax is 0.2914 with a significant level of 1%. This means that there is a positive correlation between the macro tax burden and the innovation efficiency. This finding is consistent with the conclusions in some literature. As Furman et al. [30] argued, the governments are able to invest more in the public innovation infrastructure when the macro tax burden is heavy since the governments can obtain more tax revenues. In addition, according to the logic of Shapir-Tidhar et al. [7], a higher macro tax rate will help highlight the advantages of preferential policies for innovation, which can stimulate the enthusiasm of enterprises to invest in R&D, thereby

**Table 3. Parameter estimations for the model (1).**

| Variable | (1) INN | (2) FDI | (3) INN |
|---|---|---|---|
| FDI | – | – | 0.0161** (0.0075) |
| tax | 0.2914*** (0.0748) | -1.3563*** (0.3677) | 0.3132*** (0.0709) |
| edu | 0.0898 0.2542 | 1.2200 (1.2497) | 0.0702 (0.2386) |
| inter | -0.0345 (0.0212) | -0.0977 (0.1042) | -0.0330* (0.0199) |
| indu | 0.1096* (0.0608) | 0.5604* (0.2990) | 0.1006* (0.0572) |
| urba | 0.2821** (0.1182) | 1.2570* (0.5813) | 0.2619** (0.1114) |
| trad | 0.0132 (0.0113) | -0.0281 (0.0566) | 0.0136 (0.0106) |
| WGI | 0.0258 (0.0377) | 48.4150** (18.5474) | 0.0180 (0.0356) |
| GNIpc | 0.0397** (0.0178) | -0.1496* (0.0873) | 0.0421** (0.0167) |
| C | -0.1119 (0.1919) | 25.39 (94.34) | -0.1160 (0.1801) |
| Individual effect | Y | Y | Y |
| Time effect | Y | Y | Y |
| Sobel's test | | p = 0.0638 | |
| $R^2$ | 0.9449 | | |
| Estimation method | LSDV | SEM | |

Notes: Standard errors are in brackets.

***, **, and * indicate significant levels at 1%, 5%, and 10%.

accelerating the improvement of national innovation efficiency. Atanassov and Liu [57]) argue that inclusive tax cuts may lead to less government spending on public goods (e.g., research, education) and fewer positive spillovers to firms, which in turn inhibit firms' innovation output and thus negatively impact innovation efficiency. Therefore, raising the tax rate in this case should be a better option. Likewise, Mukherjee et al. [58] reject the argument that increasing corporate income tax would reduce innovation efficiency by discouraging corporate innovation. In terms of the control variables, we find that the variables *indu*, *urba*, and *GNIpc* have positive and significant estimated coefficients. That means increasing the levels of industrialization, urbanization, and economic development can contribute to the innovation efficiency.

The columns (2) and (3) of Table 3 report the estimated coefficients of the second and the third equations of model (1). The coefficients of the variable tax are -1.3563 and 0.3132 in the columns (2) and (3), respectively, and both of them are statistically highly significant. Meanwhile, the coefficient of FDI is positive and significant at the 5% significance level in the column (3). Those results indicate that there is a partial mediation effect. The macro tax burden can impose a negative impact on the FDI inflow, thereby adversely affecting innovation efficiency. This finding is consistent with Hoghogi et al. [9]. After controlling for the mediator (i.e. *FDI*), the direct effect of the macro tax burden on the innovation efficiency is 0.3132 and significant at the 1% significance level. Since the coefficient (0.3132) of tax is greater than the

corresponding coefficient (0.2914) of tax in the baseline regression in column (1), meaning that *FDI* will weaken the promoting effect of the macro tax burden on the innovation efficiency.

The coefficient of FDI is 0.0161 and significant at the 5% level in column (3) of Table 3. This shows that FDI has a positive innovation spillover effect, and it is an important source of innovation drive for all countries under the condition of open economy. FDI not only contributes to the economic growth and employment of the host country, but also plays an important role in improving innovation ability of enterprises in the host country by driving the research and development of emerging technologies and supply chain integration. Goel [59] finds the same conclusion.

The coefficient of the variable urba is 0.2619 and statistically significant at the 5% level, indicating that increasing the urbanization level can promote the innovative efficiency in sample countries. Generally, innovation activities have characteristics of spatial cluster, and the majority of patents comes from the metropolitan areas [60]. Urbanization also benefits a country's innovation output [37, 51, 52]. The coefficient of the variable *FDI* is 0.0161 and statistically significant, indicating that *FDI* inflow can contribute to the innovation efficiency. The estimate of the variable inter is negative and statistically significant at the 10% level, meaning that the internet penetration is negatively correlated with the innovation efficiency, which is partially consistent with the finding of Xiong et al. [45]. They find that the Internet has an inverted *U*-shaped impact on innovation output, that is, if the Internet penetration rate exceeds a certain level, it will restrict innovation output. The estimated coefficient of GNIpc is positive and statistically significant at the 5% level, suggesting the increase in economic development level is positively correlated with innovation efficiency. Since financial constraints are major barriers to innovative activities by businesses [54–56]. A higher level of economic development can better overcome financial constraints on innovation investment and generate more demand for innovation. The coefficient of the variable indu is positive and significant statistically at the 10% level, suggesting increasing the proportion of secondary industry value-added to *GDP* can promote the innovation efficiency. The coefficients of the variables *edu*, *trad*, and *WGI* are positive, but insignificant statistically, indicating that human capital, the trade openness and world governance index have no impact on the innovation efficiency.

Next, we use the Sobel test to test the mediation effect of *FDI*. The Sobel test is commonly used to test if the relationship between the independent variable (*X*) and dependent variable (*Y*) is mediated by a third variable (*M*). In other words, Sobel test examines whether including a mediator (*M*) in the regression analysis substantially reduces the impact of the independent variable on the dependent variable. Table 4 reports the result of the Sobel test. The *z* statics of the Sobel's test is -1.854 with a p-value of 0.064, it is not significant at the 5% level. Therefore, the mediation is partial. Specifically, the standardized indirect effect of tax is -0.011, the direct

**Table 4. Significance testing of indirect effect (standardized).**

| Estimate | Sobel |
|---|---|
| Indirect effect | -0.0114* |
| Std. Err. | 0.0062 |
| z-value | -1.8537 |
| p-value | 0.0638 |
| 95% Conf. Interval | [-0.0235, 0.0007] |

Notes: Standard errors are in brackets.

***, **, and * indicate significant levels at 1%, 5%, and 10%.

effect is 0.164, and the total effect is 0.153. That is, a standard deviation change in the macro tax burden can directly cause a 0.164 standard deviation change in the innovation efficiency. However, since the indirect effect of tax through *FDI* on the innovation efficiency is -0.011, thereby weakening the positive effect of the macro tax burden on the innovation efficiency.

## The results of the mediation effect model (2)

In order to test the non-linear effect of *FDI* on innovation efficiency conditional on macro tax burden, we estimate the third equation of model (2) using the Hansen (1999)'s panel threshold estimator. Columns (1) and (2) of Table 5 report the estimated results. The threshold effect testing shows that there is a single threshold in the threshold variable tax, and the critical value is 25.28% with a 5% significant level. When the macro tax burden is greater than 25.28%, the coefficient of *FDI* is 0.0988, which is significant at the 1% level. While the macro tax burden is less than 25.28%, the corresponding coefficient of *FDI* is -0.0031, and insignificant statistically. As a result, the indirect effect of tax on the innovation efficiency is positive but statistically insignificant when the macro tax rate is less than the critical value (25.28%), while its indirect effect is negative and statistically significant at the 5% level when the macro tax rate is greater than the critical value (see Table 6). This means that when the macro tax rate is at a low level,

**Table 5. Parameter estimations for the model (2).**

| Variable | (1) INN | (2) FDI | (3) INN |
|---|---|---|---|
| $FDI(tax \leq 25.28\%)$ FDI $FDI(tax > 25.28\%)$ | – | -0.0031 (0.0086) 0.0988*** (0.0160) | -0.1583 *** (0.0434) |
| FDI_squared | – | – | 0.0437 ** (0.0199) |
| tax | -1.3563*** (0.3677) | 0.1971** (0.0760) | -0.3925*** (0.0789) |
| edu | 1.2200 (1.2497) | 0.0124 (0.2531) | -0.7931** (0.3817) |
| inter | -0.0977 (0.1042) | -0.1374*** (0.0160) | -0.1625*** (0.0284) |
| indu | 0.5604* (0.2990) | 0.0616 (0.0603) | 0.2339*** (0.0571) |
| urba | 1.2570* (0.5813) | 0.0453 (0.1176) | 0.2683*** (0.0276) |
| trad | -0.0281 (0.0566) | -0.0137 (0.0108) | -0.0185*** (0.0067) |
| WGI | 48.4150** (18.5474) | 0.1296 (3.0317) | 0.1388*** (0.0243) |
| GNIpc | -0.1496* (0.0873) | -0.0100 (0.0157) | -0.0647*** (0.0134) |
| C | 25.39 (94.34) | 0.6707*** (0.1567) | 1.1172*** (0.1214) |
| Individual effect | Y | | |
| Time effect | Y | | |
| One threshold | | 25.28% | |
| Threshold variable | | tax | |

Notes: Standard errors are in brackets.

***, **, and * indicate significant levels at 1%, 5%, and 10%.

**Table 6. Parameter estimations for the model (2).**

| Estimate | Sobel (when $tax \geq 25.28\%$ | Sobel (when $tax < 25.28\%$) |
|---|---|---|
| Indirect effect | -0.0694*** | 0.0019 |
| Std. Err. | 0.0222 | 0.0062 |
| z-value | -3.1218 | 0.3048 |
| p-value | 0.0018 | 0.7605 |
| 95% Conf. Interval | [-0.0258, -0.1130] | [-0.0102, 0.0140] |

Notes: Standard errors are in brackets.

***, **, and * indicate significant levels at 1%, 5%, and 10%.

the macro tax burden does not influence the innovation efficiency indirectly through *FDI*, but when the macro tax burden is at a high level, it indirectly hinders innovation efficiency by inhibiting *FDI* inflows.

## Discussion

The estimated coefficients of *FDI* in the column (2) of Table 5 indicate that there is a nonlinear relationship between the variables *FDI* and *INN*. In order to test the robustness of this nonlinear relationship, we specify the following model:

$$INN_{it} = \lambda_1 FDI_{it} + \lambda_2 FDI_{it}^2 + \alpha' tax_{it} + \theta_n C_{it}^n + \epsilon_{it} \tag{4}$$

We estimate the model (4) by using two-way fixed effect estimator. The column (3) of Table 5 displays the estimated coefficients. As expected, the coefficient of *FDI* is negative and significant at the 1% level, and the coefficient of the term of *FDI_squared* is positive and statistically significant, indicating there is a *U*-shaped relationship between FDI and *INN*. This result also confirms the robustness of the estimates of model (2) with threshold effect.

## Conclusion

Relevant literature shows that macro tax burdens have a negative impact on national innovation efficiency, because high tax burdens will inhibit the innovative activities of innovative entities. This paper uses a dataset from 54 countries to examine the impact of macro tax burden on national innovation efficiency based on the mediation effect model. In particular, based on the mediation effect model with threshold effect, this paper examines the indirect effect of macro tax burden on the innovation efficiency through FDI. The relevant conclusions are as follows:

First, the macro tax burden is positively correlated with the innovation efficiency. Though some studies find that the tax burden has a negative effect on innovation, we find that a higher macro tax rate is correlated with a higher innovation efficiency. In our mediation effect model estimations, the direct effect of macro tax burden on innovation efficiency is 0.313, the indirect effect is -0.022, and the total effect is 0.291. This means that a 1% increase in the macro tax rate can directly lead to a 0.313% increase in the innovation efficiency, and indirectly contribute to a 0.022% decrease in the innovation efficiency by inhibiting FDI inflows. As a result, increasing the macro tax rate by 1% will finally increase the innovation efficiency by 0.29%. The main reason is that if a nation has a higher macro tax rate, its government can collect more revenues, so it can invest more money in the public innovation infrastructure, which can in turn contribute to the innovation efficiency.

Second, there is a non-linear effect of FDI on innovation efficiency conditional on macro tax rate. We find that when the macro tax rate is less than 25.28%, the macro tax burden does not influence the innovation efficiency indirectly through FDI, but when the macro tax burden is greater than 25.28%, it indirectly limits innovation efficiency by hindering FDI inflows. The reason is that though FDI inflows generally can promote innovation efficiency through spillovers, this effect will change due to changes in the macro tax rate. When macro tax rate is at a high level, it will limit FDI inflows and its spillovers, and finally restrict the improvement of innovation efficiency.

Third, although it is generally believed that low tax rates are conducive to the innovative activities of micro-entities, and therefore most people advocate reducing taxes to stimulate innovative behavior of enterprises, our research shows that lower macro tax burdens are not better. In order to promote innovation efficiency at the national level, the macro tax burden should be maintained at a high level. Under this situation, the government can raise more revenues, thereby being able to invest more in national innovation infrastructure, and ultimately promote the continuous improvement of innovation efficiency.

## Author Contributions

**Conceptualization:** Xiaobo Shen, Pingsheng Dai.

**Data curation:** Pingsheng Dai.

**Formal analysis:** Xiaobo Shen, Pingsheng Dai.

**Methodology:** Xiaobo Shen, Pingsheng Dai.

**Project administration:** Pingsheng Dai.

**Resources:** Pingsheng Dai.

**Software:** Pingsheng Dai.

**Supervision:** Xiaobo Shen, Pingsheng Dai.

**Validation:** Xiaobo Shen.

**Visualization:** Pingsheng Dai.

**Writing – original draft:** Pingsheng Dai.

**Writing – review & editing:** Xiaobo Shen, Pingsheng Dai.

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
