## [Decision Letter · Decision Letter 0]

13 Aug 2024

PONE-D-24-14404Macro tax burden, FDI, and national innovation efficiency: A study on the impact of macro tax burden on national innovation efficiencyPLOS ONE

Dear Dr. Dai,

Thank you for submitting your manuscript to PLOS ONE. After careful consideration, we feel that it has merit but does not fully meet PLOS ONE’s publication criteria as it currently stands. Therefore, we invite you to submit a revised version of the manuscript that addresses the points raised during the review process.

We look forward to receiving your revised manuscript.

Kind regards,

Mosi Rosenboim

Academic Editor

PLOS ONE

2.Please review your reference list to ensure that it is complete and correct. If you have cited papers that have been retracted, please include the rationale for doing so in the manuscript text, or remove these references and replace them with relevant current references. Any changes to the reference list should be mentioned in the rebuttal letter that accompanies your revised manuscript. If you need to cite a retracted article, indicate the article’s retracted status in the References list and also include a citation and full reference for the retraction notice.

Additional Editor Comments:

Now I received reports from two judges. Please review the suggestions of both judges and correct the comments accordingly. Then I will return the revised article back to them for further review. Good Luck

Reviewers' comments:

Reviewer's Responses to Questions

**Comments to the Author**

1. Is the manuscript technically sound, and do the data support the conclusions?

Reviewer #1: Yes

Reviewer #2: Yes

2. Has the statistical analysis been performed appropriately and rigorously? 

Reviewer #1: Yes

Reviewer #2: Yes

3. Have the authors made all data underlying the findings in their manuscript fully available?

Reviewer #1: Yes

Reviewer #2: Yes

4. Is the manuscript presented in an intelligible fashion and written in standard English?

Reviewer #1: Yes

Reviewer #2: Yes

5. Review Comments to the Author

Reviewer #1: This is an interesting paper and a very decent data base. I enjoyed reading the work. I have the following few comments

1. The contribution of the paper should be better highlighted in the introduction section.

2. The literature is overall well synthesized, but i think the gap in the literature should come out more strongly

3, I appreciate the analysis of the mediation effect, but i think the paper will benefit more from a discussion of this effect from a still deeper theoretical perspective. Same for the threshold effect

4. The limitations of using SEM should also be discussed

5 The findings could also be better discussed in related to the literature

6 lastly the implications of the findings, both for theory and policy , merit more considerations

Reviewer #2: Introduction

“a too low tax burden not only weakens the government revenue, but also dilutes innovation incentives of the government’s tax credit policies” p.1 – Citation is required.

Models

When measuring the critical macro tax, one should consider the TSE index as well (see Shapir-Tidhar, M. H., Malul, M., & Rosenboim, M., 2023). I’d recommend adding the TSE index to the control variables, if feasible, to determine whether it affects the results.

Variables

In your model, you use the technological efficiency of innovation activity as a measurement of innovation efficiency (INN). Please refer to this measurement in your introduction. Is this measurement accepted in the literature? If so, please cite relevant studies. If not – please explain your rationale for choosing this measurement, and why you use this translog production (Again: Is it accepted in the literature? If so, please provide a citation for an article using this production. If not, please explain why you chose this production).

Please clarify how you estimated the INN index from the TE. You wrote “calculate the innovation efficiency index (INN) according to the estimated coefficients and inefficiency values.” (p. 4). The transition between the TE variable and the INN is not so clear.

Data

In Table 2 (p. 6), please provide the names of the countries associated with the max/min values for each variable.

Results:

“In addition, a higher macro tax rate will help highlight the advantages of preferential policies for innovation, which can stimulate the enthusiasm of enterprises to invest in R&D, thereby accelerating the improvement of national innovation efficiency” (p.7) – please add a citation.

Conclusion:

“and indirectly contribute to a -0.022% increase in the innovation efficiency” (p. 10) – the sentence is confusing and seems contradictory. Is it increase or decrease?

6. PLOS authors have the option to publish the peer review history of their article (what does this mean?). If published, this will include your full peer review and any attached files.

Reviewer #1: No

Reviewer #2: No

---

## [Author Response · Author response to Decision Letter 0]

28 Aug 2024

Review Comments to the Author

Reviewer #1: This is an interesting paper and a very decent data base. I enjoyed reading the work. I have the following few comments:

1. The contribution of the paper should be better highlighted in the introduction section.

2. The literature is overall well synthesized, but I think the gap in the literature should come out more strongly.

Response: thanks a lot for these two valuable comments. We have added the contributions of this paper and discussed the gap in the literature in the introduction section. Please see the colored text on pages 2.

3. I appreciate the analysis of the mediation effect, but i think the paper will benefit more from a discussion of this effect from a still deeper theoretical perspective. Same for the threshold effect.

Response：Thank you very much for this suggestion. We have theoretically discussed the mediation effect of FDI and the threshold effect of tax burden in the revised manuscript (please see the colored text on page 4).

4. The limitations of using SEM should also be discussed.

Response: thanks a lot for this good advice. We have discussed the limitations of SEM in the colored text on page 2.

5. The findings could also be better discussed in related to the literature

Response: thank you very much for this suggestion. We discuss the main findings of this paper in a comparison with those of the existing literature. Please see the colored text on pages 7 and 8.

6. Lastly the implications of the findings, both for theory and policy, merit more considerations

Response: Thank you very much. We have discussed the theoretical and policy implications of our findings. Please see the colored text on page 2 (paragraph 1).

Reviewer #2

Introduction

“a too low tax burden not only weakens the government revenue, but also dilutes innovation incentives of the government’s tax credit policies” p.1 – Citation is required.

Response: thank you for this advice. We have added a citation for this statement (please see the reference “ Shapir-Tidhar et al., 2023” on page 1.

Models

When measuring the critical macro tax, one should consider the TSE index as well (see Shapir-Tidhar, M. H., Malul, M., & Rosenboim, M., 2023). I’d recommend adding the TSE index to the control variables, if feasible, to determine whether it affects the results.

Response: Thank you very much for the valuable suggestion. Measuring the TSE Index requires access to categorical tax revenues, for example, in order to calculate the TSE index, Shapir-Tidhar et al. (2023) uses six categories of tax revenues from the OECD’s annual National Accounts. Unfortunately, we are unable to obtain the necessary data to calculate the TSE index for our sample countries (most of them are non-OECD member countries), so it is infeasible to add TSE into our model as a control variable. We hope we can try it in our future relevant research.

Variables

In your model, you use the technological efficiency of innovation activity as a measurement of innovation efficiency (INN). Please refer to this measurement in your introduction. Is this measurement accepted in the literature? If so, please cite relevant studies. If not – please explain your rationale for choosing this measurement, and why you use this translog production (Again: Is it accepted in the literature? If so, please provide a citation for an article using this production. If not, please explain why you chose this production).

Response: thank you very much for this pertinent advice. We have referred to the technological efficiency of innovation activity as a measurement of innovation efficiency in the introduction section. This measurement is common in the literature, we have cited a relevant study*. Similarly, using the translog production function form is very popular in literature when measuring the technological efficiency. We also have cited a relevant study (Gonzalez-Blanco et al., 2024. please see line 8 on page 2).

* Zeng JY, Ribeiro-Soriano D, Ren J. Innovation efficiency: a bibliometric review and future research agenda. Asia Pacific Business Review. 2021, 27(2):209-228.

Please clarify how you estimated the INN index from the TE. You wrote “calculate the innovation efficiency index (INN) according to the estimated coefficients and inefficiency values.” (p. 4). The transition between the TE variable and the INN is not so clear.

Response: Thank you very much for this good advice. Actually, we made a mistake in model (3) in the manscript. The lnTEit should be lnYit, and Yit is a measure of innovation output (measured by the number of granted patents). After estimating the model (3), we can calculates the TE by using inefficiency term of model (3) according to formula TE = exp(- ). As we have stated, the technological efficiency of innovation activity (TE) is a measurement of innovation efficiency (INN).

Data

In Table 2 (p. 6), please provide the names of the countries associated with the max/min values for each variable.

Response: Thank you very much for this advice. We have added the names of the countries as required.

Results:

“In addition, a higher macro tax rate will help highlight the advantages of preferential policies for innovation, which can stimulate the enthusiasm of enterprises to invest in R&D, thereby accelerating the improvement of national innovation efficiency” (p.7) – please add a citation.

Response: Thanks a lot for this advice. We have provided a citation in the revised manuscript (please see line 9 of the first paragraph on page 7).

Conclusion:

“and indirectly contribute to a -0.022% increase in the innovation efficiency” (p. 10) – the sentence is confusing and seems contradictory. Is it increase or decrease?

Response: thank you very much for this comment. This sentence is confusing indeed. We have expressed this sentence as follows: “and indirectly contribute to a 0.022% decrease in the innovation efficiency by inhibiting FDI inflows”. Hope it is more clearly (please see the colored sentence on page 11).

---

## [Decision Letter · Decision Letter 1]

8 Oct 2024

Macro tax burden, FDI, and national innovation efficiency: A study on the impact of macro tax burden on national innovation efficiency

PONE-D-24-14404R1

Dear Dr. Dai,

We’re pleased to inform you that your manuscript has been judged scientifically suitable for publication and will be formally accepted for publication once it meets all outstanding technical requirements.

Kind regards,

Mosi Rosenboim

Academic Editor

PLOS ONE

Additional Editor Comments (optional):

Reviewers' comments:

Reviewer's Responses to Questions

**Comments to the Author**

1. If the authors have adequately addressed your comments raised in a previous round of review and you feel that this manuscript is now acceptable for publication, you may indicate that here to bypass the “Comments to the Author” section, enter your conflict of interest statement in the “Confidential to Editor” section, and submit your "Accept" recommendation.

Reviewer #1: All comments have been addressed

Reviewer #2: All comments have been addressed

2. Is the manuscript technically sound, and do the data support the conclusions?

Reviewer #1: Yes

Reviewer #2: Yes

3. Has the statistical analysis been performed appropriately and rigorously? 

Reviewer #1: Yes

Reviewer #2: Yes

4. Have the authors made all data underlying the findings in their manuscript fully available?

Reviewer #1: Yes

Reviewer #2: Yes

5. Is the manuscript presented in an intelligible fashion and written in standard English?

Reviewer #1: Yes

Reviewer #2: Yes

6. Review Comments to the Author

Reviewer #1: I am overall happy with this version of the paper , Most of the comments have been adequately addressed

Reviewer #2: (No Response)

7. PLOS authors have the option to publish the peer review history of their article (what does this mean?). If published, this will include your full peer review and any attached files.

Reviewer #1: No

Reviewer #2: No

---

## [Editor Report · Acceptance letter]

11 Oct 2024

PONE-D-24-14404R1 

PLOS ONE

Dear Dr. Dai, 

I'm pleased to inform you that your manuscript has been deemed suitable for publication in PLOS ONE. Congratulations! Your manuscript is now being handed over to our production team.

Kind regards, 

on behalf of

Dr. Mosi Rosenboim 

Academic Editor

PLOS ONE